# Suspected Suicide Attempt and Intentional Misuse Cases Aged 50+ Involving Amphetamine or Methylphenidate and Medical Outcomes: Associations with Co-Used Other Substances

**Namkee G. Choi [1],* , Bryan Y. Choi [2] and S. David Baker [3]**

[1]  Steve Hicks School of Social Work, The University of Texas at Austin, Austin, TX 78712, USA
[2]  Department of Emergency Medicine, Philadelphia College of Osteopathic Medicine & Bayhealth Medical Center, Dover, DE 19901, USA; bchoiemdoc@gmail.com
[3]  Central Texas Poison Center, Temple, TX 76508, USA; david.baker@bswhealth.org
*   Correspondence: nchoi@austin.utexas.edu

**Abstract:** Introduction: Given the increasing adult use of amphetamine and methylphenidate and their high misuse potential, we examined suspected suicide attempts and other intentional misuse and medical outcomes and their associations with co-used other substances among amphetamine and methylphenidate exposure cases aged 50+. Methods: Using the 2015–2021 U.S. National Poison Data System (N = 7701 amphetamine and/or methylphenidate cases), we fit two generalized linear models for a Poisson distribution with a log link function, with suspected suicide attempt versus intentional misuse and major medical effect/death versus other outcomes as the dependent variables. Results: Of all amphetamine/methylphenidate exposure cases, suspected suicide attempts and intentional misuse were 28.4% and 13.2%, respectively. Benzodiazepine use was associated with a higher likelihood, but any illicit drug use was associated with a lower likelihood of suspected suicide attempts compared to intentional misuse. The type of stimulant involved (amphetamine or methylphenidate) was not significant. The co-use of antidepressants (IRR = 1.43, 95% CI = 1.16–1.76), prescription opioids (IRR = 1.48, 95% CI = 1.21–1.81), drugs for cardiovascular disease (IRR = 1.51, 95% CI = 1.20–1.90), antipsychotics (IRR = 1.26, 95% CI = 1.02–1.55), or illicit drugs (IRR = 2.40, 95% CI = 1.82–3.15) was associated with a higher likelihood of major effect/death. Conclusions: Suspected suicide attempts or intentional misuse accounted for more than 40% of amphetamine or methylphenidate exposure cases aged 50+. The higher likelihood of major effect/death in cases involving antidepressants, antipsychotics, and cardiovascular disease drugs also suggests the confounding effects of comorbid mental and physical health problems. Careful monitoring of those who were prescribed amphetamine or methylphenidate and use other substances is needed.

**Keywords:** amphetamine; methylphenidate; suicide attempt; intentional misuse; older adults

## 1. Introduction

There has been a significant increase over the past decade in adult use of amphetamine (common brand name: Adderall) and methylphenidate (common brand name: Ritalin), both central nervous system stimulants, especially among women and the 45+ age group [1,2]. Amphetamine and methylphenidate are primarily prescribed for narcolepsy and attention deficit hyperactivity disorder [3–7]. Off-label use of amphetamine and methylphenidate has also shown therapeutic potential for late-life, post-stroke depression and rehabilitation, depression in medically ill patients, Parkinson's disease, age-related cognitive decline (e.g., apathy in dementia), catatonia, fall prevention (e.g., gait and postural instability), and anorexia [8]. However, amphetamine and methylphenidate can have serious side effects, and clinical trials of amphetamine and methylphenidate have shown high attrition rates due to adverse effects [6,9,10]. For example, amphetamine and methylphenidate

can increase blood pressure, induce arrhythmias, and contribute to the development of cardiomyopathies and sudden death [11–13].

Amphetamine and methylphenidate are also classified as Schedule II drugs by the U.S. Controlled Substances Act given their high potential for nonmedical use and misuse/abuse (referred to as misuse hereafter) despite having an accepted medical use [14,15]. Intentional misuse (e.g., for performance enhancement, recreational use, and use in amounts or with methods that are harmful) of amphetamine and methylphenidate, including clandestinely produced amphetamine and diverted amphetamine and methylphenidate, has steadily increased over the years [16–18]. Negative physiological, cognitive, and behavioral effects on individuals who misuse amphetamine and methylphenidate are serious public health issues [19]. Between 2008 and 2015, amphetamine-related hospitalizations increased to a greater degree than hospitalizations associated with other substances, and in-hospital mortality was higher for amphetamine-related than other hospitalizations [20]. A study of the 2012–2017 U.S. National Poison Data System also found that compared to unintentional oral exposure to amphetamine, all nonmedical (including suicide attempt) amphetamine exposure cases via intravenous injection, inhalation, and intentional ingestion were at a greater risk for critical care admissions and adverse medical outcomes, including death [21].

In all age groups, intentional misuse of amphetamine or methylphenidate has been associated with other substance use and use disorders [22–29]. Studies have also found that attention deficit and hyperactivity disorder often co-occurs with substance use disorders and is associated with the early onset and more severe development of substance use disorders and with reduced treatment effectiveness [30–32]. These previous studies indicate the importance of examining other substance use/misuse associated with suspected suicide attempts and other intentional misuse and medical outcomes among adult amphetamine or methylphenidate users. In particular, despite the high suicide attempt rate among intentional amphetamine misusers [21], little research has been done on these suicide attempts and co-used other substances in such attempts.

In the present study, based on the 2015–2021 U.S. National Poison Data System's amphetamine and methylphenidate cases aged 50+, our primary research aims were to examine associations between co-used other substances and (1) suspected suicide attempts versus other intentional misuses, and (2) major medical outcomes (major effect/death) in suspected suicide attempt and other intentional misuse cases, controlling for demographic variables. Our focus on cases aged 50+ is significant since this age group in both sexes has had the highest suicide rates in the U.S. [33]. A recent study also showed that among female suicide decedents, those aged 45+ had higher rates of poisoning as a suicide method than younger decedents (37% and 41% of the 45–64 and 65+ age groups, respectively, vs. 19% and 25% of the 18–24 and 25–44 age groups), and that those who used poisoning had a significantly higher rate of mental and substance use disorders [34]. The present study's findings will provide important insights into late-life suicide attempts and other intentional misuses involving amphetamine or methylphenidate and polysubstance use.

## 2. Materials and Methods

### 2.1. Data Source

The National Poison Data System (NPDS) includes data from 55 poison control centers in the U.S. (See the NPDS website (https://aapcc.org/data-system, accessed on 1 March 2023) or Gummin et al. [35] for detailed descriptions.) Although the NPDS lists cases, not individuals, the extent to which these cases include duplicate individuals is minimal as poison center specialists are trained to detect duplication and correct it as soon as it is discovered. In this study, we focused on amphetamine and/or methylphenidate exposure (i.e., for all reasons except withdrawal and bites/stings) cases aged 50+ that were closed between 1 January 2015 and 31 December 2021. Cases were identified by substance/product ID/generic category codes and involving any number of other substances. The 7701 cases thus identified included those with all associated medical outcomes, except indirectly reported deaths (*n* = 48 from Arizona and 2 from all other states). Indirectly reported

deaths (that poison control centers acquired from medical examiners or media but did not manage [35]) have been sporadic, and Arizona's high number was due to the inclusion of deaths from state vital statistics from 2017 through 2021. The cases were from all 50 states, the District of Columbia, Puerto Rico, and unknown/refused geographic areas. Based on the authors' institutional review board guidelines, an institutional review board exemption was assumed for the analysis of these de-identified data.

### 2.2. Measures

*Exposures involving amphetamine and methylphenidate:* Based on the NPDS substance codes, we identified "amphetamine and related compounds" and "methylphenidate" exposure cases. The small number of cases ($n = 36$) that involved both amphetamine and methylphenidate were treated as amphetamine cases based on our preliminary analysis finding that these cases were more similar to amphetamine than methylphenidate cases in terms of exposure reasons and medical outcomes. Although the NPDS provides the substance sequence numbers for each case, we included all amphetamine and methylphenidate cases regardless of the sequence number as these numbers in many cases tend to be non-systematically assigned and do not signify substance priority.

*Exposure reasons*: the NPDS listed the following: unintentional (therapeutic error, adverse reaction, other unintentional misuse, or exposure via environmental/other routes); suspected suicide attempt; intentional misuse (including intentional but unknown reasons); malicious intent by others; and unknown reasons.

*Medical outcomes*: The NPDS has the following medical outcome categories for human exposure: no effect; minor effect; moderate effect; major effect; death; no follow-up, judged to be nontoxic (clinical effects not expected); no follow-up, minimal/no more than minor clinical effects possible; unable to follow, judged to be a potentially toxic exposure; and indirectly reported death. In this study, we combined these outcomes into two categories: major effect/death ($n = 592$ major effects and $n = 71$ deaths) vs. all others (except indirectly reported deaths that were excluded from this study). As opposed to moderate effect referring to "signs or symptoms that were not life-threatening or had no residual disability or disfigurement," major effect refers to "signs or symptoms that were life-threatening or resulted in significant residual disability or disfigurement" [35].

*Co-used other substances* included benzodiazepines; antidepressants; any of 23 types of NPDS-coded prescription opioids; drugs for cardiovascular diseases; antipsychotics; antihistamines; muscle relaxants; alcoholic beverages; marijuana; methadone; and illicit drugs (cocaine, methamphetamine, heroin, illicit fentanyl, phencyclidine, or lysergic acid diethylamide). Codes for illicit fentanyl and analogs were added to the NPDS on 30 October 2019.

*The covariates* in multivariable models included exposure year (2015–2021), U.S. census region, age group (50–59 and 60+ or 50–59, 60–69, and 70+ years), and sex.

*For descriptive purposes*, we reported the route of administration (ingestion, inhalation, or injection) management/care site (on-site (non-healthcare facility), treated/evaluated/released from a healthcare facility, admitted to a psychiatric facility, admitted to a noncritical care unit; admitted to a critical care unit; refused referral/did not arrive at healthcare facility/lost to follow-up/left against medical advice), and the number of all substances involved.

### 2.3. Analysis

All analyses were conducted with Stata 17/MP (Stata Corp, College Station, TX, USA). We first reported the changes in amphetamine and methylphenidate cases by age group (50–59 and 60+) and sex during the study period (2015–2021) for all exposure reasons. Second, we used $\chi^2$ and ANOVA to compare demographic and exposure-related characteristics, medical outcomes, and co-used substances among three exposure reason groups: unintentional exposure, suspected suicide attempt, and intentional misuse, i.e., excluding cases with unknown exposure reasons ($n = 282$) or exposure via others' malicious intent ($n = 17$). Third, to examine primary research questions, we fit two generalized

linear models for a Poisson distribution with a log link function. The first generalized linear model (GLM) focused on co-used substances in suspected suicide attempt cases versus intentional misuse cases, and the second GLM focused on co-used substances in major effects/deaths among suspected suicide attempt and intentional misuse cases. GLMs rather than logistic regression models were used as odds ratios tend to exaggerate the true relative risk when the event (i.e., attempted suicides in the study) is a common (i.e., >10%) occurrence [36]. As a preliminary diagnostic, we used the variance inflation factor, using a cut-off of 2.50 [37], from linear regression models to assess multicollinearity among covariates, which indicated no concerning multicollinearity among covariates. The GLM results are reported as incidence rate ratios (IRRs) with 95% confidence intervals (CIs). Statistical significance was set at $p < 0.05$.

## 3. Results

### 3.1. Amphetamine and Methylphenidate Cases, 2015–2021, by Age Group and Sex

Of the 7701 cases including all exposure reasons, 5416 (70.3%) were amphetamine and 2285 (29.7%) were methylphenidate cases, and two-thirds of both amphetamine (67.3%) and methylphenidate (68.1%) cases were female. Figure 1 shows that amphetamine cases increased between 2018 and 2021 for the 50–59 age group and between 2017 and 2021 for the 60+ age group. While the 50–59 age group was significantly larger than the 60+ age group throughout the seven-year study period, the 60+ age group had a proportionally greater increase over the years (Pearson $\chi^2$(df = 6) = 15.64, $p$ = 0.016). On the other hand, methylphenidate cases did not significantly change between 2015 and 2021 for either age group (Pearson $\chi^2$(df = 6) = 2.16, $p$ = 0.904).

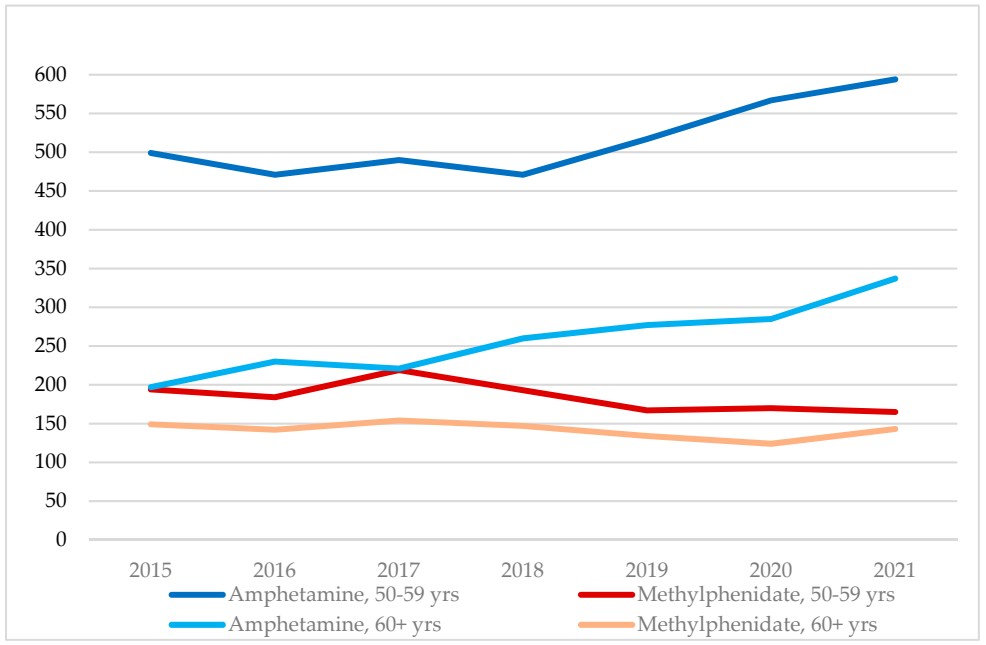

**Figure 1.** Number of amphetamine and methylphenidate exposure cases by age group, 2015–2021.

Figure 2 shows that female amphetamine cases increased to a greater extent than male amphetamine cases during the seven years (Pearson $\chi^2$(df = 6) = 19.50, $p$ = 0.003). Male amphetamine cases were stable between 2015 ($n$ = 244) and 2021 ($n$ = 253). Methylphenidate cases did not significantly change between 2015 and 2021 for both sexes (Pearson $\chi^2$(df = 6) = 10.31, $p$ = 0.113).

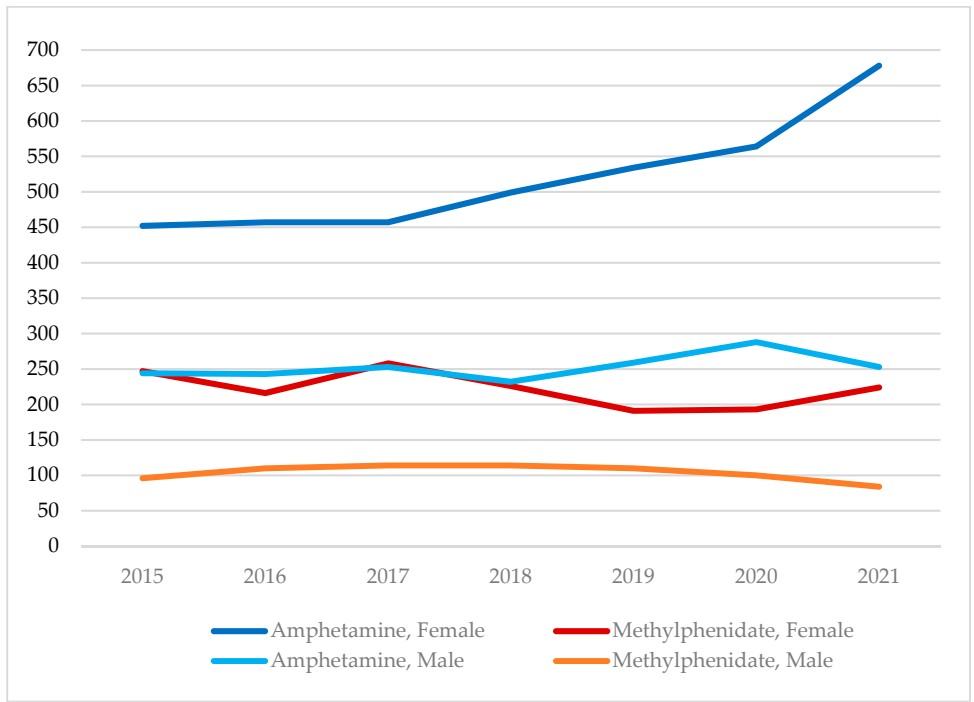

**Figure 2.** Number of amphetamine and methylphenidate exposure cases by gender, 2015–2021.

Additional analysis showed that one-half of amphetamine and 71.1% of methylphenidate cases were unintentional exposures; 30.5% of amphetamine and 19.6% of methylphenidate cases were suspected suicide attempts; and 15.0% of amphetamine and 7.3% of methylphenidate cases were intentional misuse cases. Major effect/death was reported for 10.2% of amphetamine cases and 4.8% of methylphenidate cases; however, there was no difference between amphetamine and methylphenidate cases in terms of the number of other substances involved ($M$ = 2.57 ($SD$ = 2.27) for amphetamine cases and $M$ = 2.48 ($SD$ = 2.35) for methylphenidate cases; $t$ = 1.687, $p$ = 0.092).

*3.2. Unintentional Exposure, Suspected Suicide Attempt, and Intentional Misuse*

Table 1 shows that after excluding exposures due to unknown reasons or malicious intent, unintentional exposures, suspected suicide attempts, and intentional misuse were 58.4%, 28.4%, and 13.2%, respectively, of combined amphetamine and methylphenidate cases. Unintentional exposure (e.g., inadvertently took medication twice, took medications too close together, or took someone else's medication) cases significantly differed from suspected suicide attempt and intentional misuse cases in demographic characteristics, route of administration, management/care site, other substance involvement, and medical outcomes (i.e., 0.7% major effect/death vs. 18.8% for suspected suicide attempts and 17.0% for intentional misuses).

Compared to intentional misuse cases, suspected suicide attempt cases included higher proportions of >70-year-olds, women, critical care admissions, and users of benzodiazepines (34.9% vs. 15.4%), antidepressants, antipsychotics, cardiovascular disease drugs, antihistamines, muscle relaxants, and alcohol, but included a lower proportion of marijuana and illicit drug users (3.0% vs. 12.0%). Co-used illicit drugs among intentional misuse cases included cocaine (5.9%), methamphetamine (3.9%), heroin (1.9%), illicit fentanyl (0.6%), phencyclidine (0.6%), and/or lysergic acid diethylamide (0.2%). (Illicit fentanyl was added to the NPDS substance codes on 30 October 2019.) No difference was found in the proportions that used prescription opioids and methadone and in medical outcomes.

**Table 1.** Demographic and exposure-related characteristics of amphetamine and methylphenidate cases. Age 50+, 2015–2021, by exposure reason.

| | (1) Unintentional Exposure 4325 (58.4%) | (2) Suspected Suicide 2100 (28.4%) | (3) Intentional Misuse 977 (13.2%) | P: among 3 Groups | P: between (2) and (3) |
|---|---|---|---|---|---|
| Year (%) | | | | 0.030 | 0.067 |
| 2015 | 13.2 | 12.9 | 14.6 | | |
| 2016 | 13.3 | 13.3 | 13.9 | | |
| 2017 | 13.7 | 14.2 | 15.1 | | |
| 2018 | 13.9 | 14.9 | 11.5 | | |
| 2019 | 14.0 | 14.3 | 16.1 | | |
| 2020 | 14.6 | 15.0 | 15.8 | | |
| 2021 | 17.3 | 15.3 | 13.0 | | |
| US census region (%) | | | | <0.001 | 0.416 |
| Northeast | 12.2 | 14.1 | 15.8 | | |
| Midwest | 23.5 | 26.5 | 25.4 | | |
| South | 41.9 | 40.7 | 38.5 | | |
| West | 22.0 | 18.5 | 20.4 | | |
| Puerto Rico/unknown | 0.4 | 0.1 | 0 | | |
| Age group (in years; %) | | | | <0.001 | 0.017 |
| 50–59 | 55.9 | 73.4 | 75.1 | | |
| 60–69 | 29.5 | 21.6 | 22.1 | | |
| 70+ | 14.5 | 5.0 | 2.8 | | |
| Sex (%) | | | | <0.001 | <0.001 |
| Female | 74.4 | 64.6 | 45.7 | | |
| Male | 25.6 | 35.4 | 54.3 | | |
| Route of administration (%) | | | | | |
| Ingestion | 77.5 | 85.0 | 81.6 | <0.001 | 0.020 |
| Inhalation/nasal | 12.5 | 15.5 | 13.2 | 0.004 | 0.101 |
| Injection | 2.3 | 3.2 | 3.5 | 0.031 | 0.746 |
| Management/care site (%) | | | | <0.001 | <0.001 |
| On-site (non-HCF) | 71.2 | 0 | 5.4 | | |
| Healthcare facility treated/evaluated and released | 15.2 | 16.4 | 26.6 | | |
| Admitted to a psychiatric facility | 0.4 | 21.4 | 7.5 | | |
| Admitted to a noncritical care unit | 3.0 | 19.0 | 18.6 | | |
| Admitted to a critical care unit | 1.8 | 38.7 | 31.6 | | |
| No-show/lost to follow-up/left against medical advice/unknown | 8.4 | 4.5 | 10.2 | | |
| No. of substances involved, M (SD) | 2.09 (2.14) | 3.51 (2.45) | 2.38 (1.82) | <0.001 | <0.001 |
| Other substances used (%) | | | | | |
| Benzodiazepine | 3.5 | 34.9 | 15.4 | <0.001 | <0.001 |
| Antidepressants | 8.9 | 19.4 | 7.4 | <0.001 | <0.001 |
| Prescription opioids | 2.7 | 18.2 | 16.5 | <0.001 | 0.264 |
| Drugs for cardiovascular disease | 13.1 | 13.7 | 4.8 | <0.001 | <0.001 |
| Antipsychotics | 5.8 | 19.6 | 8.2 | <0.001 | <0.001 |
| Antihistamine | 3.0 | 6.5 | 3.8 | <0.001 | 0.002 |
| Muscle relaxant | 1.4 | 7.8 | 4.6 | <0.001 | 0.002 |
| Alcohol | 0.5 | 18.1 | 10.6 | <0.001 | <0.001 |
| Marijuana | 0.1 | 2.5 | 4.4 | <0.001 | 0.007 |
| Methadone | 0.3 | 1.2 | 1.7 | <0.001 | 0.243 |
| Illicit drug [1] | 0.2 | 3.0 | 12.0 | <0.001 | <0.001 |
| Medical outcomes (%) | | | | | |
| Major effect/death | 0.7 | 18.8 | 17.0 | <0.001 | 0.229 |
| All other | 99.3 | 81.2 | 83.0 | | |

[1] Including cocaine, methamphetamine, heroin, lysergic acid diethylamide, phencyclidine, and illicit fentanyl; codes for illicit fentanyl and analogs were added to the NPDS on 30 October 2019. Note: Probability values were calculated based on Pearson's $\chi^2$ tests or ANOVA. Cases with unknown exposure reasons ($n = 282$) or exposure via others' malicious intent ($n = 17$) were excluded from the analyses.

### 3.3. Associations between Co-Used Substances and Suspected Suicide Attempts vs. Intentional Misuses

The first column of Table 2 shows that benzodiazepine use (IRR = 1.09, 95% CI = 1.06–1.16) was associated with a higher likelihood but any illicit drug use (IRR = 0.83, 95% CI = 0.73–0.95) was associated with a lower likelihood of suspected suicide attempts compared to intentional misuse. The type of stimulant involved (amphetamine or methylphenidate) was not significant. Of the covariates, only female sex (IRR = 1.07, 95% CI = 1.01–1.13) was associated with a higher likelihood of suspected suicide attempts.

**Table 2.** Associations of suspected suicide attempt vs. intentional misuse and of major effect/death vs. other outcomes with co-used other substances: generalized linear modeling results.

| | Suspected Suicide vs. Intentional Misuse/Abuse | Major Effect/Death vs. All Other Outcomes |
|---|---|---|
| | IRR (95% CI) | IRR (95% CI) |
| Suspected suicide: vs. Intentional misuse/abuse | | 1.06 (0.87–1.28) |
| Amphetamine vs. Methylphenidate | 0.97 (0.91–1.04) | 1.24 (0.99–1.56) |
| Benzodiazepine | 1.09 (1.03–1.16) ** | 1.17 (0.98–1.41) |
| Antidepressants | 1.07 (0.99–1.16) | 1.43 (1.16–1.76) ** |
| Prescription opioids | 1.00 (0.93–1.08) | 1.48 (1.21–1.82) *** |
| Drugs for cardiovascular disease | 1.07 (0.90–1.13) | 1.51 (1.20–1.90) ** |
| Antipsychotics | 1.06 (0.98–1.14) | 1.26 (1.02–1.55) * |
| Antihistamine | 1.01 (0.90–1.13) | 0.87 (0.61–1.24) |
| Muscle relaxant | 1.01 (0.91–1.13) | 1.13 (0.84–1.52) |
| Alcohol | 1.06 (0.99–1.14) | 0.87 (0.68–1.11) |
| Marijuana | 0.97 (0.82–1.15) | 1.14 (0.75–1.73) |
| Methadone | 0.97 (0.76–1.25) | 1.19 (0.68–2.08) |
| Any illicit drug | 0.83 (0.73–0.95) ** | 2.40 (1.82–3.15) *** |
| Year: vs. 2015 | | |
|    2016 | 1.03 (0.92–1.14) | 1.00 (0.71–1.42) |
|    2017 | 1.02 (0.91–1.13) | 1.08 (0.78–1.50) |
|    2018 | 1.05 (0.95–1.17) | 1.19 (0.85–1.65) |
|    2019 | 1.01 (0.91–1.12) | 1.44 (1.05–1.97) * |
|    2020 | 1.03 (0.93–1.14) | 1.27 (0.93–1.75) |
|    2021 | 1.05 (0.95–1.16) | 1.13 (0.82–1.57) |
| US census region [1]: vs. South | | |
|    Northeast | 0.98 (0.90–1.06) | 1.43 (1.11–1.84) ** |
|    Midwest | 1.00 (0.93–1.07) | 1.50 (1.22–1.84) *** |
|    West | 1.00 (0.93–1.08) | 1.13 (0.88–1.44) |
| Age: vs. 50–59 years | | |
|    60–69 years | 1.00 (0.93–1.07) | 1.10 (0.90–1.34) |
|    70+ years | 1.05 (0.92–1.20) | 1.25 (0.85–1.83) |
| Female vs. Male | 1.07 (1.01–1.13) * | 1.10 (0.92–1.31) |
| N | 3072 | 3072 |

[1] Cases in Puerto Rico or unknown locations were excluded from both models. * $p < 0.05$; ** $p < 0.01$; *** $p < 0.001$.

*3.4. Associations between Co-Used Substances and Major Effect/Death among Suspected Suicide Attempt and Intentional Misuse Cases*

The second column of Table 2 shows that the co-use of antidepressants (IRR = 1.43, 95% CI = 1.16–1.76), prescription opioids (IRR = 1.48, 95% CI = 1.21–1.82), cardiovascular disease drugs (IRR = 1.51, 95% CI = 1.20–1.90), and antipsychotics (IRR = 1.26, 95% CI = 1.02–1.55) or illicit drugs (IRR = 2.40, 95% CI = 1.82–3.15) was associated with a higher likelihood of major effect/death. Whether the exposure was a suspected suicide attempt or intentional misuse was not a significant factor. Of the covariates, the year 2019, compared to 2015, and the Northeast and Midwest regions, compared to the South region, were associated with a higher likelihood of major effect/death.

Illicit drug co-use was associated with both suspected suicide attempts and major effects/death. It was associated with a lower likelihood of suspected suicide attempts but a higher likelihood of major effects/death. Figure 3 shows adjusted predicted rates of suspected suicide attempts and major effects/death among cases involving any illicit drug.

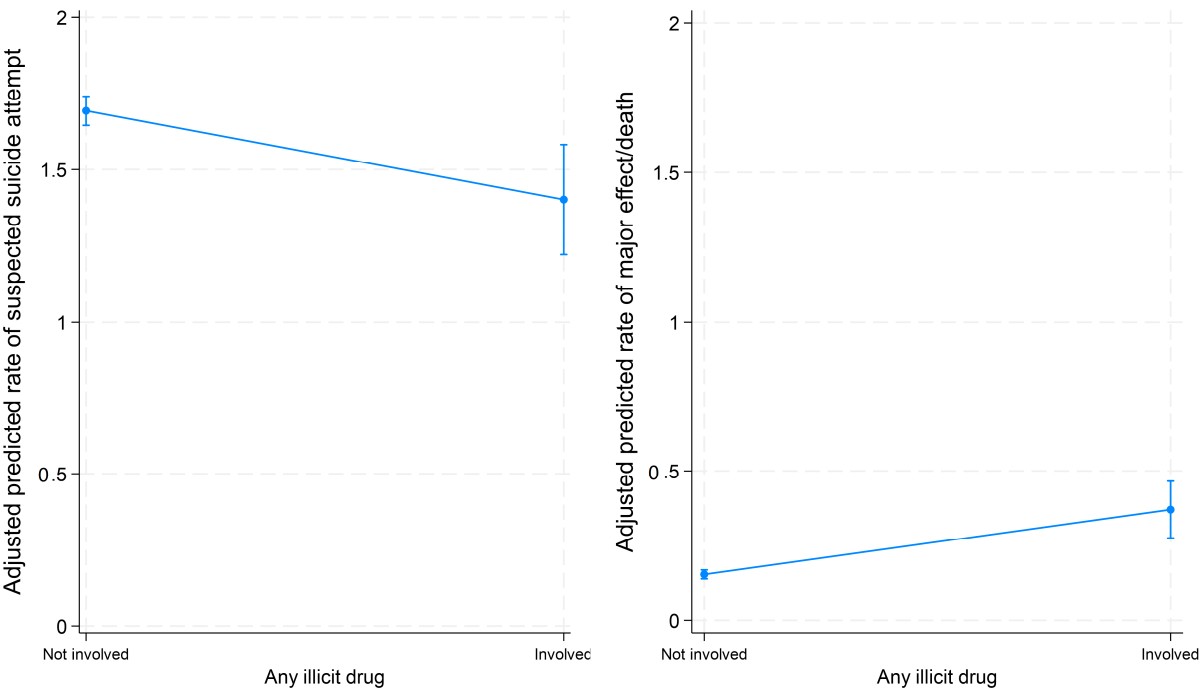

**Figure 3.** Adjusted predicted rates of suspected suicide attempts and major effects/death with 95% CIs among cases involving amphetamine or methylphenidate by use of any illicit drug, 2015–2021.

## 4. Discussion

### 4.1. Summary and Contributions

We examined co-used substances in suspected suicide attempts or other intentional misuses in amphetamine or methylphenidate cases aged 50+ in the NPDS. While nonmedical use and use disorders of prescription stimulants are lower than those of prescription opioids or sedatives [38,39], it is important to examine polysubstance use in amphetamine and methylphenidate cases given increases in these cases in recent years. Our findings showed that amphetamine cases increased between 2017/2018 and 2021, especially among the 60+ age group and women. Nearly 42% of amphetamine and methylphenidate cases were suspected suicide attempts or other intentional misuses, and as expected, the co-use of other substances and major effects/death were significantly higher among these cases than among unintentional exposure cases.

Multivariable findings show that among suicide attempt and other intentional misuse cases, benzodiazepine co-use was associated with a higher likelihood of suspected suicide attempts, and illicit drug co-use was associated with a higher likelihood of other inten-

tional misuses. Multivariable findings also show that the significant correlates of major effect/death are prescription opioids, antidepressants, antipsychotics, cardiovascular disease drugs, and illicit drugs, without any significant difference between suspected suicides and intentional misuses or between amphetamine and methylphenidate.

The higher likelihood of benzodiazepine co-use in suspected suicide attempts compared to other intentional misuses appears to be consistent with previous findings. An analysis of the censuses of emergency department and inpatient discharges for suicidal intentional overdoses in 11 US states found that 19.6–22.5% of the suicidal overdoses involved benzodiazepines, and 15.4–17.3% involved opioids [40]. However, the study found that benzodiazepines were most often involved in nonfatal acts, while opioids were most commonly identified in fatal suicide poisonings among adults, followed by barbiturates, antidepressants, antidiabetics, calcium channel blockers, alcohol, and psychostimulants [40].

The significant associations of major effects/death with co-used prescription opioids and illicit drugs in suspected suicide attempt and intentional misuse cases are also consistent with previous research findings. For example, a study found significantly more 12-month emergency department visits and hospitalizations, controlling for fair/poor health and other characteristics, among older adult prescription opioid users, compared to nonusers [41]. Although the rate of illicit drug use and use disorders is, in general, lower among those aged 50+ than among younger adults [39], the effects of these substances are more severe because of aging-related changes in pharmacokinetics and drug metabolism [42,43]. Serious health consequences from illicit drug use include adverse cardiovascular, renal, and cognitive effects, inflammation, and overdose deaths [44–48].

The higher likelihood of major effects/death in suspected suicide attempt and intentional misuse cases involving antidepressants, antipsychotics, and cardiovascular disease drugs also shows the likely confounding effects of comorbid mental and physical health problems in these cases. An analysis of the 2013–2018 U.S. Medical Expenditure Survey found a marked increase in the risk of amphetamine and methylphenidate misuse in a population often reporting multiple neurological or mental disorders and taking medications for depression and anxiety [1]. A study also found a significant association between stimulant exposure in early life and earlier onset of psychosis [49]. We also speculate that the co-use of amphetamine or methylphenidate and antipsychotics and cardiovascular disease drugs may have been necessary to treat side effects (e.g., psychotic symptoms and adverse cardiovascular events) of amphetamine or methylphenidate [50]. Future research is needed to examine this possibility.

Of the covariates, a higher likelihood of suspected suicide attempts among female than male amphetamine or methylphenidate users may stem from the fact that poisoning is more often used as a suicide method among women than men [34]. However, the reasons for the 44% higher likelihood of major effects/death in 2019 than in 2015 but not in other years are not clear. The reasons for significantly higher rates of major effects/death in the Northeast and Midwest regions than in the South region are not clear, either, although the regional differences are in line with the higher drug overdose death rates in the Northeast and Midwest than in the South [51].

### 4.2. Study Limitations

The study has the following limitations due to data constraints: First, since the NPDS contains only exposures that were reported to poison control centers, they do not represent all exposures among the population, limiting the findings' generalizability. The increase in amphetamine cases among the 60+ age group and women between 2017/2018 and 2021 may reflect a rise in poisoning incidents in these population groups; however, it may also be a result of better reporting practices. Second, deaths among cases are underestimated as not all cases were followed up. Moreover, it is not clear if all reported deaths were related to substance use or were due to other causes. Third, data that are telephone-reported to poison centers without medical record validation and toxicological confirmation may compromise validity. This applies to suspected suicide attempts as it is unclear whether or not the

exposure reason was self-reported by the individual or verified/reported by any health care provider. Fourth, many NPDS cases had missing data on medication dose and route of administration; thus, these variables could not be included in our analysis. Fifth, the lack of data on characteristics such as race, pre-existing health conditions, health insurance, and substance use history also precluded more detailed analyses of sociodemographic and health-related factors.

*4.3. Clinical Implications*

Despite these limitations, the study has the following implications: First, with increasing amphetamine exposure cases in older adults, healthcare professionals need to monitor adverse events and unintentional or intentional misuse. They need to pay special attention to increased risks of suicide attempts, especially among older women, and adverse medical outcomes when amphetamine or methylphenidate are co-used with other psychotropic drugs, prescription opioids, or cardiovascular disease drugs. Second, although a small proportion of amphetamine and methylphenidate cases co-used illicit drugs, the significant association between major effects/death and illicit drug use points to the importance of screening these drugs. For cases referred to and/or admitted to a critical care unit, substance use and mental health treatment programs need to be an important treatment component.

## 5. Conclusions

Amphetamine and methylphenidate misuse in the 50+ age group may be lower than other prescription drug misuses/abuses; however, the increase in amphetamine cases reported to poison control centers in recent years is a warning sign that preventive measures are needed to stem that increase, especially the suicide attempt and intentional misuse cases. Careful monitoring of those who were prescribed amphetamine and methylphenidate as well as continued research on nonpharmacological treatment of attention deficit hyperactivity disorder, narcolepsy, and other conditions are needed. Those who misuse amphetamine or methylphenidate also need to be provided easy access to integrated physical and behavioral health services.

**Author Contributions:** Conceptualization, N.G.C., B.Y.C. and S.D.B.; methodology, N.G.C., B.Y.C. and S.D.B.; software, N.G.C.; validation, N.G.C., B.Y.C. and S.D.B.; formal analysis, N.G.C.; investigation, N.G.C., B.Y.C. and S.D.B.; resources, N.G.C. and S.D.B.; data curation, N.G.C. and S.D.B.; writing—original draft preparation, N.G.C. and B.Y.C.; writing—review and editing, N.G.C., B.Y.C. and S.D.B.; visualization, N.G.C.; supervision, N.G.C. and S.D.B.; project administration, N.G.C. and S.D.B.; funding acquisition, N.G.C. All authors have read and agreed to the published version of the manuscript.

**Funding:** This research received no external funding.

**Institutional Review Board Statement:** The American Association of Poison Control Centers (AAPCC) maintains the National Poison Data System (NPDS), which houses de-identified case records of self-reported information collected from callers during exposure management and poison information calls managed by the country's poison control centers (PCCs). NPDS data do not reflect the entire universe of exposures to a particular substance as additional exposures may go unreported to PCCs; accordingly, NPDS data should not be construed to represent the complete incidence of U.S. exposures to any substance(s). Exposures do not necessarily represent a poisoning or overdose and AAPCC is not able to completely verify the accuracy of every report. Findings based on NPDS data do not necessarily reflect the opinions of the AAPCC.

**Informed Consent Statement:** Patient consent was waived due to the fact that this study was a secondary, deidentified data analysis.

**Data Availability Statement:** The authors were granted access to NPDS based on the American Association of Poison Control Centers review committee's review of our proposal. The authors are not allowed to share the data set with unauthorized people. Requests to access NPDS should be directed to https://aapcc.org/data-system (accessed on 1 March 2023).

**Acknowledgments:** The American Association of Poison Control Centers made the National Poison Data System (NPDS) available to the authors for this study. This study's findings and conclusions are those of the authors alone and do not necessarily represent the official position of the American Association of Poison Control Centers or participating poison control centers.

**Conflicts of Interest:** The authors declare no conflict of interest.

**Abbreviations**

| | |
|---|---|
| NPDS | The National Poison Data System |
| GLM | Generalized linear model |
| IRR | Incidence rate ratio |
| CIs | Confidence intervals |

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
