# Peer review of "Suspected Suicide Attempt and Intentional Misuse Cases Aged 50+ Involving Amphetamine or Methylphenidate and Medical Outcomes: Associations with Co-Used Other Substances"

_2813-0618, doi:10.3390/pharma2030020_

Round 1

Reviewer 1 Report

The manuscript concerns an extremely important and current issue, which is suicide attempts under the influence of selected psychoactive substances. The layout of the study is well-thought-out, typical for articles. The introduction is written clearly and clearly. It guides the reader into the topic, it is a perfect background for the rest of the article. The purpose of the work is written precisely and clearly. The population analyzed in the work is impressive. The results were presented in a correct and interesting way. The discussion of the results is comprehensive. Ethically, work without reservations. The bibliography used is appropriate to the topic discussed.

Author Response

Dear Reviewer #1,

Thank you for taking time to review our paper and providing such wonderful endorsement of our study. We truly appreciate your positive review.

Comments and Suggestions for Authors:

The manuscript concerns an extremely important and current issue, which is suicide attempts under the influence of selected psychoactive substances. The layout of the study is well-thought-out, typical for articles. The introduction is written clearly and clearly. It guides the reader into the topic, it is a perfect background for the rest of the article. The purpose of the work is written precisely and clearly. The population analyzed in the work is impressive. The results were presented in a correct and interesting way. The discussion of the results is comprehensive. Ethically, work without reservations. The bibliography used is appropriate to the topic discussed.

Submission Date

03 May 2023

Date of this review

18 May 2023 14:07:07

Reviewer 2 Report

Dear Authors;

I found this work an interesting examination of the  suspected suicide attempts and other intentional misuse and  medical outcomes and their associations with co-used other substances among amphetamine and  methylphenidate exposure cases age 50+. It needs some extra work prior to further processing it. Regards. P.S.

[1] Writing:

1-1 Abbreviations: Add list of used abbreviations in manuscript at the end of text before Reference section. Example:  Abbreviations: GLM: General Linear Model; ....

1-2 Missing Section and Subsection Numbers: Add these to the work. It is hard to follow the structure of the manuscript.

1-3 References: Make sure they are in mdpi format. Example: Years for papers are in bold format, etc

1-4 Discussion: Hard to follow it up. Break it down it with something like this: 4.Discussion; 4.1. Summary & Contributions; 4.2. Strengths & Limitations; 4.3. Future Work

1-5 Consort Diagram: Add Consort Diagram with exclusion criteria at each stage of analysis to subsection "2.1. Data Sources". 

[2] Statistical:

2-1 Missing Values: Explain how did you address the missing cases in the analysis. Explain in the manuscript revision.

2-2 Interaction Terms in GLM models: Did you check them ? Explain in the manuscript revision.

2-3 Missing Plot: The main message of the paper is missing through showcasing it by a plot. Referring to page 7-8( Table 2), add a plot with main variable of interest on X axis and IRR(95% CI) on Y axis split for several categories of X axis. Show the trend and discuss it in the manuscript revision.  See Some Examples Below:

Sample Links: 

2-3-1: https://www.biostat.jhsph.edu/courses/bio622/misc/graphci_methods_2009_revised.pdf

2-2-2:

https://blog.uvm.edu/tbplante/2018/03/14/code-to-make-a-dot-and-95-confidence-interval-figure-in-stata/ 

Author Response

Dear Reviewer #2,

Thank you for taking time to review our paper and providing very helpful suggestions. We have incorporated all your suggestions, except one for which we provided our justifications. Please see summary of our revision below. Major text revisions are shown using track changes. Again, we are grateful for your review.

Dear Authors;

I found this work an interesting examination of the suspected suicide attempts and other intentional misuse and medical outcomes and their associations with co-used other substances among amphetamine and methylphenidate exposure cases age 50+. It needs some extra work prior to further processing it. Regards. P.S.

RESPONSE: Thank you for your helpful suggestions for improving our study.  

[1] Writing:

1-1 Abbreviations: Add list of used abbreviations in manuscript at the end of text before Reference section. Example:  Abbreviations: GLM: General Linear Model; ....

RESPONSE:  Done. Thanks for the suggestion.

1-2 Missing Section and Subsection Numbers: Add these to the work. It is hard to follow the structure of the manuscript.

RESPONSE: Done.  Thank you for the suggestion.

1-3 References: Make sure they are in mdpi format. Example: Years for papers are in bold format, etc

RESPONSE: Done. (The revised reference list is shown with track changes accepted.)

1-4 Discussion: Hard to follow it up. Break it down it with something like this: 4.Discussion; 4.1. Summary & Contributions; 4.2. Strengths & Limitations; 4.3. Future Work

RESPONSE: Done. We now have 4.1. Summary & Contributions; 4.2. Study Limitations; 4.3. Clinical Implications

1-5 Consort Diagram: Add Consort Diagram with exclusion criteria at each stage of analysis to subsection "2.1. Data Sources". 

RESPONSE: Thanks for this suggestion. However, since we are not reporting a clinical trial, we believe that our description of case selections in the Data Source section is sufficient.

[2] Statistical:

2-1 Missing Values: Explain how did you address the missing cases in the analysis. Explain in the manuscript revision.

RESPONSE: NPDS does not have “missing cases” as it includes only those reported to the Poison Control Centers. However, a significant proportion of cases had missing data for some variables, e.g., medication dose and route of administration. We now included the following sentence in the Limitations section: “Fourth, many NPDS cases had missing data on medication dose and route of administration; thus, these variables could not be included in our analysis.”  In the Analysis and Results sections, we also clarified that 299 cases with unknown exposure reasons or exposure vis others’ malicious intent were excluded in our main analyses. 

2-2 Interaction Terms in GLM models: Did you check them ? Explain in the manuscript revision.

RESPONSE: We did not include interaction analysis as it was not part of our research questions.

2-3 Missing Plot: The main message of the paper is missing through showcasing it by a plot. Referring to page 7-8 (Table 2), add a plot with main variable of interest on X axis and IRR (95% CI) on Y axis split for several categories of X axis. Show the trend and discuss it in the manuscript revision.  See Some Examples Below:

Sample Links: 

2-3-1: https://www.biostat.jhsph.edu/courses/bio622/misc/graphci_methods_2009_revised.pdf

2-2-2:

https://blog.uvm.edu/tbplante/2018/03/14/code-to-make-a-dot-and-95-confidence-interval-figure-in-stata/ 

RESPONSE: Thanks for this recommendation and examples. We now included “Figure 3. Adjusted predicted rates of suspected suicide attempt and major effect/death with 95% CIs among cases involving amphetamine or methylphenidate by use of any illicit drug, 2015-2021.” The two graphs were plotted using the GLM results reported in Table 2. We chose to focus on illicit drugs of all co-used substances because only Illicit drug co-use was associated with both suspected suicide attempt and major effect/death. It was associated with a lower likelihood of suspected suicide attempt but a higher likelihood of major effect/death.

Submission Date

03 May 2023

Date of this review

10 Jun 2023 07:09:11

Date of Authors’ revision

14 June 2023

Round 2

Reviewer 2 Report

Dear Authors, most of my comments were addressed satisfactorily. Regards.